# The Tissue Response to Hypoxia: How Therapeutic Carbon Dioxide Moves the Response toward Homeostasis and Away from Instability

**DOI:** 10.3390/ijms24065181

**Published:** 2023-03-08

**Authors:** Richard J. Rivers, Cynthia J. Meininger

**Affiliations:** 1Department of Anesthesia and Critical Care Medicine, Johns Hopkins University, Baltimore, MD 21218, USA; 2Department of Medical Physiology, Texas A&M University School of Medicine, Bryan, TX 77807, USA

**Keywords:** transdermal carbon dioxide, hypoxia, hypoxia-inducible factor-1a, angiogenesis, diabetes, peripheral vascular disease, exercise, CO_2_ therapy, carboxy therapy

## Abstract

Sustained tissue hypoxia is associated with many pathophysiological conditions, including chronic inflammation, chronic wounds, slow-healing fractures, microvascular complications of diabetes, and metastatic spread of tumors. This extended deficiency of oxygen (O_2_) in the tissue sets creates a microenvironment that supports inflammation and initiates cell survival paradigms. Elevating tissue carbon dioxide levels (CO_2_) pushes the tissue environment toward “thrive mode,” bringing increased blood flow, added O_2_, reduced inflammation, and enhanced angiogenesis. This review presents the science supporting the clinical benefits observed with the administration of therapeutic CO_2_. It also presents the current knowledge regarding the cellular and molecular mechanisms responsible for the biological effects of CO_2_ therapy. The most notable findings of the review include (a) CO_2_ activates angiogenesis not mediated by hypoxia-inducible factor 1a, (b) CO_2_ is strongly anti-inflammatory, (c) CO_2_ inhibits tumor growth and metastasis, and (d) CO_2_ can stimulate the same pathways as exercise and thereby, acts as a critical mediator in the biological response of skeletal muscle to tissue hypoxia.

## 1. Introduction

Oxygen (O_2_) is carried in the blood and delivered to cells in tissues. O_2_ is necessary for normal cell respiration, serving as an electron acceptor in the electron transport system. Carbon dioxide (CO_2_) is produced as a byproduct of cellular respiration. Normal tissue function requires a balance between O_2_ delivery and CO_2_ removal. When blood flow is too low or when tissue O_2_ demand exceeds delivery (such as during exercise), tissue hypoxia ensues, and metabolic byproducts, including CO_2_, build up in the tissues [1,2]. CO_2_ causes dilation of blood vessels, bringing about greater blood flow, increased O_2_ delivery, and removal of vasodilatory metabolites. The resulting reduction in vasodilatory signals brings blood flow back to homeostatic levels. This metabolic regulation of blood flow guarantees an adequate supply of blood to the tissues and microvascular stability. In this way, the local supply of O_2_ by the circulation is matched to the local demand by the tissues.

Cardiovascular and microvascular diseases can be caused by the chronic disruption of normal blood flow that makes previously healthy tissues cascade into a severe and, at times, permanent pathological condition. The condition caused by sustained hypoxia results in a slowed metabolism that is dependent on glycolytic pathways and a chronic inflammatory state that is perpetuated by an increase in hypoxia-inducible factor 1α (HIF-1α) production. Multiple transcriptional responses to HIF-1α promote a survival response in cells, preventing cell death in response to continued hypoxia [3]. Beyond cell survival, a major function of HIF-1α is the transcriptional activation of genes involved in new blood vessel growth. This neovascularization improves blood flow to supply more O_2_ to the tissue and reverse the hypoxia.

In contrast with the survival mode induced by severe or chronic hypoxia, the limited or transient hypoxia brought about by increased metabolic demand in tissues results in an acute decrease in ATP stores that is sensed by cells via the action of adenosine monophosphate-activated protein kinase (AMPK) [4]. AMPK senses cellular energy status by monitoring the ATP:AMP ratio. When this ratio is decreased, AMPK restores energy balance by inhibiting anabolic reactions that consume ATP and promoting catabolic processes that generate ATP. AMPK activation in vascular endothelial cells results in vasodilation and new blood vessel growth to increase blood flow and oxygen delivery to the tissue, thus restoring normal tissue function [4].

CO_2_ levels in the tissue rise immediately in response to increased metabolic demand. Elevated tissue CO_2_ acts as a sensitive trigger for cellular responses to hypoxia, modulating biological signaling pathways (e.g., HIF-1α and AMPK) to restore normal tissue function and homeostasis. This review presents data highlighting the potential use of CO_2_ as a therapeutic approach to tissue hypoxia to promote tissue repair, reduce inflammation, reverse the pathophysiological effects of hyperglycemia, and enhance the efficacy of cancer therapy. Moreover, it summarizes the status of knowledge of the cellular and molecular mechanisms underlying these beneficial effects of therapeutic CO_2_ delivery.

## 2. Therapeutic Effects of Carbon Dioxide

The use of CO_2_ as a therapeutic treatment for various ailments dates back hundreds of years when people visited and built cities around the hot springs of Europe. The medicinal effects of the waters found in hot springs have been attributed to dissolved CO_2_ in the form of carbonic acid [5]. There are now many more methods for using therapeutic CO_2_. Such treatment regimens used today include carbonated water (i.e., baths) (CB) [6], CO_2_ gas applied to the skin (transdermal [TD]) [7,8], intra-arterial infusions of carbonated solutions into tumors (IA) [9], intraperitoneal injection of CO_2_ (IP) [10], inhaled CO_2_ or hypercarbia (IH) [11], carbonated paste or gel (CG) [12], and subcutaneous microinjections of CO_2_ (SQ) [13]. In all these therapeutic applications, except for SQ, CO_2_ is applied for 10–20 min. Treatment schedules vary from 1–3 days apart, and the number of treatments varies from one (SQ, IA, and IP) to 4–20 for the other therapies. Dosing studies are rare, but one study showed that treatments must be applied every 2–3 days to be effective in healing fractures [14]. No studies have noted any adverse side effects or complications from CO_2_ therapy except for SQ, where bleeding or bruising can occur. It should be noted, however, that CO_2_ is anti-inflammatory, so treating an infected wound should be done with caution.

Modern studies have shown that therapeutic CO_2_ applied to wounds, tumors, and through the skin to underlying tissues has dramatic and long-lasting effects on inflammation, oxygenation, angiogenesis, and healing. Clinical and experimental findings using therapeutic CO_2_ are summarized here.

### 2.1. Accelerated Wound and Fracture Healing

There are now several studies investigating the effects of therapeutic CO_2_ on wounds and fractures. Studies in rats treated with TD CO_2_ demonstrated increased survival of skin flaps due to increased blood flow and greater capillary density [15]. This correlated with increased gene expression of basic fibroblast growth factor and vascular endothelial growth factor (VEGF) but a paradoxical decrease in HIF-1α. In a retrospective study of topical TD CO_2_ treatment of 86 human patients with chronic wounds and 17 patients with acute wounds, all patients had resolution of their wounds [16]. The CO_2_ gas was able to penetrate the granulation tissue of the wound bed to support/accelerate healing while improving the microcirculation and reducing the incidence of infection. Humans with diabetes often have chronic wounds that do not heal and eventually lead to the need for amputation. A double-blind study showed that TD CO_2_ therapy would heal these long-standing wounds [17]. By the end of the study, 66% of the CO_2_-treated wounds had healed completely, and wound volume decreased by 99%. In the control group, none of the wounds completely healed.

Clinicians have very limited options to biologically improve bone fracture repair. Blood flow and angiogenesis at the site of a fracture are critical components of fracture repair. When rats with long bone fractures were treated with TD CO_2_, fracture union was accelerated [14,18]. For example, fracture union was evident at week 3 in 86% of animals in the treated group compared with 36% in the control group [18]. Increased blood flow and greater capillary density at the fracture site were accompanied by elevated expression of chondrogenic and osteogenic genes resulting in early production of collagen and cartilage. Increased angiogenesis and blood flow induced by the TD CO_2_ treatment likely promoted blood vessel invasion and transformation of the avascular cartilaginous matrix into vascularized osseous tissue.

In the Phase I clinical trial involving 19 human patients with fractures of the lower extremities, CO_2_ was applied to the affected limb, and blood flow was measured in both limbs [19]. This TD CO_2_ therapy promoted an approximately two-fold increase in blood flow in the fractured limbs. As a result of the increase in blood flow, enhanced local oxygenation likely contributed to tissue healing. Importantly, no adverse events, including hypercapnia, were noted in these patients, supporting the validity of continuing assessments of CO_2_ therapy in additional clinical trials. Increased blood flow and greater O_2_ delivery suggest that CO_2_ therapy will be beneficial for the treatment of fractures in humans, including fractures in patients with the ischemic disease or diabetes mellitus, in smokers, and in those individuals with avascular nonunion [19].

### 2.2. Increased Blood Flow to Ischemic Limbs

Treatment of ischemia in limbs due to peripheral vascular disease is very challenging and commonly includes multiple surgeries and loss of toes and/or limbs. Experiments in mice and rats with unilateral hindlimb ischemia showed that CB and TD CO_2_ therapy induced elevated levels of VEGF in addition to activating the nitric oxide-cyclic guanosine-monophosphate (NO-cGMP) system to increase blood flow to the limb [20,21]. Treatment increased capillary density and collateral vessel formation in the ischemic limb as well as the number of endothelial cell progenitors in the circulation [21].

TD CO_2_ improved blood flow and preserved tissue in humans with peripheral vascular disease [22,23,24]. In a study of patients with ischemic ulceration due to severe stenosis or vascular occlusion, 83% of limbs were salvaged with CO_2_ therapy, thus reducing the need for amputation in patients with critical limb ischemia (CB) [22]. Transcutaneous CO_2_ acts as a vasodilator, opening nonfunctioning capillaries to enhance the O_2_ diffusion area (CB) [23]. Increases in tissue CO_2_ concentration affect both blood flow (tissue perfusion) and the capacity of hemoglobin to release O_2_ (the shift in the O_2_ dissociation curve known as the Bohr effect). Therapeutic TD CO_2_ reduced claudication (pain caused by reduced blood flow to the legs) and increased pain-free and total walking distance by 66% and 73%, respectively, in patients with critical limb ischemia [24]. Importantly, these effects were maintained for 12 months after cessation of therapy.

### 2.3. Greater Blood Flow and Vascularization in Diabetes

Diabetes is a major risk factor for the development of peripheral vascular disease. In a rat model of diabetes mellitus with hindlimb ischemia, TD therapy resulted in a two-fold increase in peak and mean blood flow in hindlimb skeletal muscle as well as a two-fold increase in the number of small blood vessels in skeletal muscle sections (CB) [6]. Both the structure and function of the microvasculature of skeletal muscle are known to be altered in diabetes. Hyperglycemia, resulting from poorly controlled diabetes, reduces skeletal muscle capillarization via an imbalance in pro- and anti-angiogenic factors [25]. TD CO_2_ therapy in rats with streptozotocin-(STZ-)-induced diabetes, a model of type I diabetes, increased blood flow in the soleus muscle, attenuated capillary rarefaction and restored the balance of angiogenic growth factors [26].

In addition to causing changes in vascular architecture and density in skeletal muscle, hyperglycemia can reduce oxidative capacity, the measure of a muscle’s maximal capacity to use O_2_, by reducing mitochondrial enzymatic activity and biogenesis [27]. The activity of citrate synthase (CS), a key mitochondrial enzyme in the tricarboxylic acid cycle, is used as an indicator of the oxidative capacity of skeletal muscle. CS activity in the soleus muscle was reduced in STZ-induced diabetic rats compared to nondiabetic rats, but cutaneous CO_2_ treatment increased CS activity in diabetic rats compared to diabetic rats without treatment and brought CS activity up to levels measured in nondiabetic control rats [27]. The cytochrome c oxidase (COX) complex catalyzes the final step in the mitochondrial electron transfer chain and is regarded as one of the major regulation sites for mitochondrial function and, thus, oxidative phosphorylation. Hyperglycemia reduced levels of COX subunit 4, essential for the assembly and respiratory function of the COX complex, while TD CO_2_ attenuated this effect [27].

Oxidative capacity is regulated by peroxisome proliferator-activated receptor-gamma co-activator-1 alpha (PGC-1α), and low oxidative capacity in diabetes correlates with decreased PGC-1α expression. CO_2_ therapy increases NO production in skeletal muscle via an increase in blood flow. NO can increase PGC-1α protein expression via activation of cGMP and promotes mitochondrial biogenesis and function. Of note, the application of TD CO_2_ not only increased oxidative capacity in skeletal muscle, but CO_2_ also improved hyperglycemia via an increase in glucose metabolism mediated by increased PGC-1α expression [27].

### 2.4. Improved Skeletal Muscle Function and Healing

Endurance (aerobic) exercise increases skeletal muscle oxidative capacity, mitochondrial biogenesis, and muscle fiber-type switching (via an increase in PGC-1α expression) while also stimulating angiogenesis (via upregulation of VEGF). This links the regulation of the consumption of O_2_ by mitochondria to the delivery of O_2_ and nutrients by the vasculature. Elevating tissue CO_2_ can cause a skeletal muscle to respond as though it has been exercising. For example, compared to nontreated rats, skeletal muscle in rats treated with CO_2_ displays an increase in mitochondria number as well as elevations in the expression of PGC-1α, VEGF, and sirtuin 1 (SIRT1) (TD) [28]. SIRT1 supports mitochondrial biogenesis and muscle fiber-type changes to increase ATP production and improve muscle performance. These data are consistent with the greater endurance [29] and more efficient muscle activity [30] noted in humans (CB) and rats (TD), respectively, following CO_2_ therapy.

TD application of CO_2_ may have therapeutic potential for muscle regeneration and strength recovery following injury. In a study of rats with a skeletal muscle injury, injured muscle fibers were completely repaired at week six in the CO_2_-treated group but only partially repaired in the untreated group [31]. Expression levels of the muscle-specific transcription factors MyoD and myogenin (involved in differentiation of myoblasts into myotubes) were increased at week two after injury in the CO_2_-treated group (signaling an acceleration of the regeneration process and differentiation of muscle cells), and significantly more capillaries were seen four weeks after injury. Angiogenesis is a prerequisite for morphological and functional healing, leading to the rebuilding of damaged vessels, re-establishment of blood flow, and restoration of the O_2_ supply to tissues. TD CO_2_ accelerates muscle injury repair by increasing O_2_ delivery as well as mitochondrial biogenesis, promoting VEGF upregulation to stimulate neovascularization and activating myogenesis to build muscle. TD CO_2_ therapy also ameliorated the atrophy of skeletal muscle caused by nerve injury or following fracture [32,33].

### 2.5. Reduced Inflammation

In addition to its regenerative effects, CO_2_ has anti-inflammatory effects. In laparoscopic rat models of sepsis, IP injection of CO_2_, but not helium or air, significantly increased survival in rats with lipopolysaccharide-induced sepsis [10]. This was accompanied by CO_2_-specific increases in plasma interleukin-10 (IL-10) levels and decreases in tumor necrosis factor-alpha (TNF-α) levels. The reduction in TNF-α levels correlated with the increase in survival and is consistent with previous demonstrations of IL-10-induced inhibition of macrophage-derived TNF-α production and suppression of nuclear factor kappa B (NF-kB) activation (responsible for the upregulation of many proinflammatory genes) as well as with studies showing that administration of recombinant IL-10 increases survival in septic animals and reduces inflammation [34,35,36,37].

The attenuation of the inflammatory response brought about by IP injection of CO_2_, including a reduction in the acute inflammatory response, was independent of peritoneal absorption of CO_2_ and subsequent systemic acidosis (IP) [38]. Similar effects were noted in a pig study investigating the use of IP CO_2_ gas vs. air during laparoscopic surgery [39]. Inflammatory responses were reduced by CO_2_, including a reduction in IL-6 release and peritoneal macrophage production of reactive O_2_ species (ROS).

Another key anti-inflammatory response to elevated CO_2_ was discovered in patients exhibiting hypercapnia (excessive CO_2_ in the bloodstream) due to inadequate respiration caused by extensive lung disease. These patients must be ventilated with minimal volumes to protect the lung from mechanical damage caused by lung stretching (ventilator-induced lung injury) so CO_2_ levels remain elevated in the blood. Interestingly, controlling the rise in CO_2_, the so-called “permissive hypercapnia”, was found to improve lung inflammation and reduce morbidity and mortality in patients with acute respiratory distress syndrome (ARDS) (IH) [11]. While hypercapnia-induced acidosis boosts cardiac output, increases peripheral perfusion, and enhances tissue hemoglobin O_2_ unloading, CO_2_ can penetrate and act directly on cells to modulate intracellular pathways leading to inflammation and oxidative stress as well as pathways linked to cell survival, proliferation, and apoptosis [11]. This has relevance for patients with coronavirus disease 2019 (COVID-19), in which the viral spike protein of severe acute respiratory syndrome coronavirus 2 (SARS-CoV-2) binds and disrupts the functioning of angiotensin-converting enzyme 2 (ACE2) on cell surfaces, leading to the activation of mitogen-activated protein kinases (MAPKs) and production of proinflammatory cytokines (e.g., interferon-gamma, IL-1β, IL-6, and TNF-α) that cause pneumonia or ARDS [40]. The anti-inflammatory and antioxidant effects of increased tissue CO_2_ could be beneficial for these patients.

### 2.6. Decreased Tumor Growth and Metastasis

Tumor hypoxia is a common feature of malignant tumors and contributes to their growth, invasiveness, and metastatic potential [41]. HIF-1α, a key transcription factor induced by tissue hypoxia, regulates the transcription of genes involved in angiogenesis, cell survival, and tumor cell invasion. A hypoxic microenvironment also induces various molecular pathways that allow tumors to become resistant to chemotherapy, such as the induction of multidrug resistance (MDR) genes and radiation therapy. Multiple studies have shown that TD CO_2_ therapy can reduce tumor growth and metastasis [42,43,44,45,46], while one in vivo study showed that IA infusion of a carbonated solution reduced tumor growth in a liver tumor model [9]. Importantly, several of these studies demonstrated that CO_2_ therapy enhances the effects of chemotherapy [9,45] and radiation therapy [44,46]. Thus, while CO_2_ treatment alone can decrease the potential size of tumors, when CO_2_ therapy is combined with chemotherapy or radiation, the response is even more effective. CO_2_ therapy increases tumor cell apoptosis [9,42,43], which reduces tumor growth. The ability of CO_2_ to increase blood flow could increase oxygenation of the tumor core, decreasing HIF-1α levels and reducing inflammation.

## 3. Mechanisms of Action of Carbon Dioxide Therapy

CO_2_ has been seen simply as a by-product of cellular respiration, although it has roles in the control of breathing, acid–base balance, and cerebral blood flow [47]. The therapeutic effects of CO_2_ noted above cannot be explained by our current dogma of CO_2_ being carried in the blood away from tissues. Many new functions for tissue CO_2_ have been revealed, emphasizing the importance of controlling CO_2_ levels in tissues. The cellular and molecular mechanisms by which CO_2_ therapy has its effects are only beginning to be elucidated.

### 3.1. CO_2_/H^+^ Concentrations and Carbonic Anhydrase Govern the Sensing of CO_2_ in the Regulation of Cellular Function

This review describes the use of therapeutic CO_2_. It should be noted, however, that many of the findings described herein rely on the hydration of CO_2_ and the production of protons to activate sensors and initiate a response. While hydration happens spontaneously under physiological conditions, it happens as much as six orders of magnitude faster in the presence of the enzyme carbonic anhydrase (CA) [48]. CO_2_ hydration via the action of CA is an important pathway for many responses. While the relative importance of protons and CA is discussed in the context of the effect of CO_2_ on O_2_ release from hemoglobin in red blood cells, there are many additional responses where hydration is noted. If not discussed, it does not mean that protons are not important. The role of protons is complicated by the fact that CO_2_ can affect protons extracellularly, independent of how it affects protons intracellularly, indicating the importance of the location of the CA. As more is discovered about CO_2_ biology, the relative significance of protons, and CA, in sensing the levels of CO_2_ will come to light.

### 3.2. Increasing Blood Flow and Tissue Oxygenation

Increasing the tissue concentration of CO_2_ increases blood flow to that tissue (CB, TD) [6,26], and increased blood flow stimulates flow-mediated dilation of blood vessels [49,50]. This vasodilator action is partly due to the CO_2_-induced production of NO by the endothelium (which acts via activation of cyclic GMP to relax the underlying smooth muscle) as it can be attenuated by inhibition of the enzyme endothelial NO synthase [21,51]. NO can also reduce arterial stiffness. Indeed, a single TD CO_2_ treatment reduced arterial stiffness and, thus, peripheral vascular resistance in hypertensive patients [8]. However, this reduction in arterial stiffness by CO_2_ is controlled by both endothelium-derived NO and endothelium-derived hyperpolarizing factor (EDHF) [52]. EDHF plays an important role in CO_2_-mediated shear stress-induced endothelium-dependent relaxation, and potassium channels, especially calcium-activated potassium (K_Ca_) channels, appear to be involved. EDHF-mediated hyperpolarization of the smooth muscle is evoked through myo-endothelial junctions and/or the accumulation of K^+^ in the intercellular space.

CO_2_-induced vasodilation can also be endothelium-independent. CO_2_ reacts with water to form carbonic acid, which is in equilibrium with HCO3^−^ and H^+^ ions. Elevating CO_2_ pushes the equilibrium toward more H^+^ ions. The resulting reduction in tissue pH inhibits contractility of the underlying smooth muscle in the blood vessel wall, leading to vasodilatation (CB) [50]. CO_2_ in exercising skeletal muscle also activates perivascular sensory nerves, which release calcitonin gene-related peptide (CGRP) (IA) [53]. CGRP causes long-lasting local vasodilatation. Thus, during prolonged exhaustive exercise, CO_2_ liberated from exercising muscle can directly stimulate sensory nerves to release CGRP, while the local drop in pericellular pH may act together with CO_2_ to further activate sensory nerves, releasing more CGRP to dilate vessels and further increase blood flow [53]. This mechanism allows the vasodilatory response to occur locally, not systemically.

CO_2_ and lactic acid are produced locally by metabolically active cells. The resultant decrease in intracellular pH provides a mechanism to balance vascular supply and metabolic activity by decreasing vascular resistance and increasing blood flow. While both CO_2_ and H^+^ individually can cause vasodilation via direct action on arterioles in skeletal muscle, they cannot conduct the spread of vasodilation throughout the microvascular network [54]. Instead, they likely work in concert to modify the effectiveness of other vasodilators. A physiological role for CO_2_ and pH in the regulation of coronary blood flow was initially proposed by Case and colleagues [55]. In an isolated heart preparation, CO_2_ evoked large increases in blood flow, presumably by generating adenosine (ADO) in the myocardium. The activation of ADO receptors on blood vessels by endogenously generated ADO contributes to the coronary vasodilation that occurs in response to an increase in IA CO_2_ and the resultant metabolic acidosis [56]. CO_2_ has also been reported to cause the release of ADO from cultured vascular endothelial cells [57].

An alternative mechanism for CO_2_-mediated increases in blood flow involves connexins and the release of ATP, which can act on ATP-sensitive potassium (K_ATP_) channels in vascular smooth muscle to cause vasodilation, contributing both to resting blood flow and vasodilator-induced increases in flow. Connexins form large-pore channels that function either as dodecameric gap junctions or hexameric hemichannels to allow the regulated movement of small molecules and ions across cell membranes. Hemichannels are a particularly important mechanism for the release of ATP into the extracellular space [58]. CO_2_ binds to a carbamylation motif in connexins Cx26, Cx30, and Cx32 and causes their hemichannels to open. Fluctuations in CO_2_ levels within the tissue alter their open probability [59]. While it has been shown that direct CO_2_-gated Cx26 hemichannel opening and subsequent release of ATP mediate an important part of neurovascular coupling [47], the physiological significance of the CO_2_ sensitivity of Cx30 and Cx32 has not yet been elucidated during exercise. Endothelial cells express Cx32, which participates in gap junctional communication [60], and endothelial cells have been shown to release ATP via connexin hemichannels [61]. It is possible that CO_2_ binding to endothelial Cx32 could explain the release of ATP into the blood in skeletal muscle during exercise, as the source of ATP remains to be defined [62]. CO_2_-mediated CGRP release and the CA-dependent production of protons could also activate K_ATP_ channels in vascular smooth muscle [63,64].

It should be noted that all the aforementioned mechanisms for CO_2_-dependent increases in blood flow are not relevant to pulmonary circulation. Pulmonary circulation is unique. The role of CO_2_ in controlling the distribution of pulmonary blood flow is small compared to the powerful effects of hypoxia and pulmonary vasoconstriction [65]. Data shows that CO_2_ can induce both vasodilation and vasoconstriction, and it appears to depend on the level of resting vascular tone [66,67]. In sharp contrast to the peripheral vasculature, where protons cause vasodilation, CO_2_ and the CA-dependent production of carbonic acid cause vasoconstriction in the pulmonary vascular network [68].

In the in vivo hamster cheek pouch, CO_2_ alters both microvessel diameter and the perivascular partial pressure of O_2_ (pO_2_), as demonstrated by using microelectrodes to measure O_2_ [69]. When the CO_2_ concentration of the superfusion solution was increased, the tissue O_2_ concentration also increased [69]. Since increased O_2_ tensions cause vasoconstriction, there may be a limit to how much blood flow can increase as the CO_2_ rises. This may explain the limited increases seen in cremaster blood flow when CO_2_ in the superfusion solution was elevated above 10% [54].

Finally, it has been shown in the human forearm that there is enhanced O_2_ release from hemoglobin when CO_2_ is applied to the skin (TD) [70]. This CO_2_-based elevation in tissue oxygenation can be explained by the Bohr effect, a rightward shift of the O_2_–hemoglobin dissociation curve with an increase in pCO_2_ or a decrease in pH [71]. This relationship was discovered in 1904 by Christian Bohr [71], who showed that increased CO_2_ tensions decreased the O_2_ affinity of whole blood in dogs. It should be noted, however, that Malte et al. showed that the protons generated by CA have a larger effect on the binding of O_2_ to hemoglobin than does CO_2_ itself (Figure 1) [72].

There are two Bohr groups on each hemoglobin moiety, allowing a total of eight protons to interact with the hemoglobin molecule. In the red blood cell, CO_2_ is rapidly interconverted to HCO3^−^ and H^+^ ions via CA to generate the protons that can cause this effect. In the absence of CO_2_ and protons, hemoglobin will bind O_2_ until it reaches tissue that is essentially anoxic. Thus, tissues with low metabolism or which may be exposed to the body surface will have low CO_2_ levels and, therefore, will receive little O_2_ from the blood that may be flowing through the tissue. Elevating CO_2_ to generate protons will force the rightward shift in curves so that O_2_ will be released from hemoglobin and delivered to the tissue [Graph A]. The rightward shift caused by CO_2_ alone is relatively limited.

When external CO_2_ is applied, the CO_2_ is hydrated in the presence of CA, and higher concentrations of protons are produced. The rapid hydration of CO_2_ maintains the driving force for the diffusion of CO_2_ into the red blood cell and prevents the loss of CO_2_ by conversion to HCO3^−^. The effect of protons on oxygen binding is dramatic and decreases the binding of the oxygen, as defined in Graph B. When hemoglobin binds protons during the release of oxygen, it acts as a buffer. pH will actually increase, so red cell pH is not a good indicator of proton binding to hemoglobin. According to Malte and Lykkeboe, “The Bohr effect is exerted by protons preloaded on the Bohr groups at the given pH, as well as the protons taken up by the Bohr groups during gas exchange. The proton preload sets the oxygen affinity (i.e., the P_50_) at the beginning of the gas exchange. The protons taken up during gas exchange lead to a further progressive decrease in oxygen affinity. The concerted effect of these two processes is very large, and the size of the Bohr effect has a direct influence on the oxygen affinity of hemoglobin.” [72].

Hartmann et al. [23] proposed that an increase in tissue O_2_ pressure caused by bathing with CO_2_-enriched water was due to the Bohr effect, but they were unable to show this directly. Sakai et al. [70] utilized near-infrared spectroscopy to track dynamic changes in tissue oxy- and deoxyhemoglobin levels in real-time, directly demonstrating that TD CO_2_ application causes O_2_ dissociation from oxy-hemoglobin, i.e., the Bohr effect, in humans.

### 3.3. Enhancing Angiogenesis

While a single treatment with CO_2_ can stimulate vasodilation and enhance O_2_ release from hemoglobin due to the Bohr effect, repeated CO_2_ applications maintain the tissue O_2_ supply and induce angiogenesis [6,15,18,26,28,32,33,73]. Responses to CO_2_ include increases in basic fibroblast growth factor [2,8], VEGF [6,15,18,26,28,32,33,73], SIRT1 [28], PGC-1α [16,17,20,22], and NO [26,73], all linked to neovascularization. Angiogenesis is a critical part of the healing process and is known to occur in response to tissue hypoxia. Hypoxia induces the transcription of angiogenic genes, such as VEGF, in part by the stabilization of hypoxia-inducible transcription factors [74]. A deficiency in blood flow leads to hypoxia, and elevated levels of CO_2_ accumulate in the tissue.

Interestingly, while the therapeutic introduction of CO_2_ causes angiogenesis, it decreases or prevents the production of HIF-1α [9,12,15,42,43,44,45,46,75,76]. One mechanism for CO_2_-mediated suppression of HIF-1α is via a pH-dependent, O_2_-independent degradation of HIF-1α [75]. Selfridge et al. proposed that elevated CO_2_ promotes lysosomal degradation of HIF-α subunits in an environment of decreasing intracellular pH rather than induction of proteasomal degradation [75]. Galganska et al. showed that a single treatment with CO_2_ efficiently inhibited HIF-1α expression in cultured human endothelial cells [40]. Alternatively, the decrease in HIF-1α could simply be the result of the increased oxygenation generated by the increased blood flow and increased release of oxygen because of the Bohr effect, as described above. Thus, while CO_2_ is a potent trigger for angiogenesis linked to hypoxia, it appears to raise VEGF levels via a non-HIF1 pathway.

Similarly, a non-HIF-mediated process may occur in skeletal muscle in response to exercise. Exercise generates high concentrations of CO_2_ in the skeletal muscle, and CO_2_ stimulates PGC-1α [77,78]. PGC-1α is a metabolic sensor induced by hypoxia that regulates VEGF expression and angiogenesis in cultured muscle cells and skeletal muscle in vivo [79]. PGC-1α promotes angiogenesis through interaction with estrogen-related receptor-alpha (ERRα), an orphan nuclear receptor that binds to the promoter region and a novel enhancer in the first intron of the VEGF gene to initiate robust VEGF gene activation in a HIF-independent manner [79,80,81]. A reduced increase in VEGF expression in PGC-1α knockout mice following exercise demonstrates the functional relevance of this non-HIF-1α pathway in coordinating the angiogenic response to exercise [80].

In contrast, Sopariwala et al. demonstrated that ERRα is a hypoxia-stimulated factor that activates a paracrine angiogenic gene program to promote normal as well as ischemic angiogenesis in skeletal muscle in a HIF-1-dependent manner [82]. In endothelial cells, ERRα receptors were positively linked to angiogenesis, as downregulation of ERRα receptors decreased angiogenesis [83]. This reduction in angiogenesis was mediated by inhibition of HIF-1α expression as well as inhibition of the phosphatidyl inositol-3-kinase/protein kinase B (Akt)/signal transducer and activator of transcription (PI3K/Akt/STAT3) pathway. Likhite et al., however, reported that ERRα receptors act as negative regulators of angiogenesis by transcriptionally repressing angiogenic genes [84]. The link between angiogenesis and exercise/ischemia is indeed complex, and further experimentation will be needed to elucidate the role of ERRα receptors.

The CO_2_ sensor triggering the angiogenic process has yet to be defined. Connexins have recently been shown to bind CO_2_ [58]. Cx32 is expressed on endothelial cells and regulates angiogenesis by enhancing tube formation and cell migration [85]. It is conceivable that CO_2_ binding to Cx32 initiates neovascularization in response to chronic tissue hypoxia. An alternate candidate for a tissue CO_2_ sensor is ubiquitin. While much of the control of angiogenic events is instigated through hypoxia-induced VEGF expression, the ubiquitin–proteasome system plays a central role in fine-tuning the functions of core proangiogenic proteins, including VEGF, VEGF receptors, angiogenic signaling proteins, and other non-VEGF pathways [86]. Ubiquitin is a highly conserved protein that regulates both protein activity and protein degradation through conjugation to target proteins [86,87]. CO_2_ binds to lysine residues in ubiquitin, forming carbamate, and this reversible carbamylation reaction may be involved in the diverse physiological responses to fluctuating pCO_2_ levels. Linthwaite et al. described multiple locations where CO_2_ could bind to ubiquitin and alter its properties [87]. Deng et al. described how the deubiquitination of AMPK is critical for its activation [88]. AMPK is a crucial sensor of cellular energy, and it is activated when intracellular ATP concentrations decrease, and AMP or ADP concentrations increase in response to energy or pathological stresses [89]. AMPK activation plays a critical role in the activation of PGC-1α and, thereby, could be the link between CO_2_ and exercise to initiate non-HIF-mediated angiogenesis [81].

### 3.4. Stimulating Skeletal Muscle Mitochondrial Biogenesis

Mitochondria are critical for aerobic ATP synthesis and proper cell function. Mitochondrial quantity and quality in skeletal muscle are not only important for performance but also relevant to health, as mitochondrial dysfunction is associated with muscle atrophy, diabetes, and aging. Mitochondrial biogenesis, an increase in mitochondrial number and function, is instrumental in exercise training-induced improvement of muscle function but is also important for conferring cytoprotection against several damaging stimuli and for maintaining whole-body metabolic homeostasis. As noted above, for angiogenesis, the molecular mechanisms triggered by therapeutic CO_2_ appear to mimic the mechanisms induced by exercise. Consistent with the known effects of PGC-1α, TD CO_2_, similar to exercise, increases mitochondrial biogenesis [28,30]. Oxidative and metabolic stresses induced by contractile activity in skeletal muscle stimulate PGC-1α expression and activity, which in turn promotes mitochondrial biogenesis through interactions with nuclear transcription factors nuclear respiratory factor 1 (NRF1) and NRF2 [81]. The coactivation of NRF1 and NRF2 by PGC-1α also induces mitochondrial transcription factor A (TFAM), which regulates mitochondrial DNA transcription [81]. PGC-1α thus coordinates the expression of both nuclear- and mitochondrial-encoded genes. As noted above, the exact mechanism(s) by which CO_2_, exercise, and contractile activity bring about activation of PGC-1α are not known, although p38γ3 MAPK, an upstream stress-activated kinase for PGC-1α, may have a role in activating mitochondrial biogenesis [81]. AMPK phosphorylates PCG-1a on several sites, leading to increased activation, while SIRT1 activates PCG-1a directly by removing acetylation groups on 13 lysines [90].

### 3.5. Reducing Inflammation

Some of the beneficial effects of therapeutic CO_2_ may be related to its ability to suppress inflammatory signaling and elevated ROS. Hypoxia, HIF-1α, and increased expression of glycolytic enzymes are linked to inflammation, and NF-κB can drive increased HIF-1α expression. Loss of oxygen supply, depletion of energy, and increased oxidative stress in ischemia/hypoxia result in the reduction of ATP synthesis and initiate a cascade of pathways that lead to cell death if not reversed. SIRT1 plays a major role in protecting against cellular stress and in controlling metabolic pathways during ischemia/hypoxia.

CO_2_ can elicit a specific repertoire of transcriptional events in a dose-dependent fashion [91]. NF-κB is the master regulator of the sensing and signaling pathways that induce the transcription of multiple proinflammatory genes. As described in the studies noted above and others [10,91], CO_2_ can counteract inflammation to restore homeostasis. CO_2_ disrupts inflammation by binding to ubiquitin. Once CO_2_ is bound, ubiquitin will remodel and transform into a proteasome that will directly diminish the NF-κB response [87,92]. During deubiquitination, ubiquitin is removed from a substrate to release its activity. This has been described as CO_2_-induced de-activation of NF-κB [92].

Alternatively, studies in human cells demonstrated that CO_2_ directly inhibited the activation of the MAPKs extracellular signal-regulated kinases 1 and 2 (ERK1/2) by the receptor-binding domain of the SARS-CoV-2 spike protein, independent of pH and independent of upstream activators, blocking the rise in proinflammatory cytokine production. Thus, ERK1/2 act as a direct CO_2_ sensor, mediating the anti-inflammatory response of elevated CO_2_ [40].

### 3.6. Antioxidant Effects

The presence of CO_2_ is essential for life due to its antioxidant effects. While CB CO_2_ can decrease inflammation, it will also reduce the ROS generated during the various phases of inflammation [93,94,95]. As Veselá and Wilhelm noted [95], CO_2_ prevents nitration and oxidative damage by scavenging peroxynitrite. Moreover, CO_2_ can protect superoxide dismutase from hydrogen-peroxide-induced damage, although this does result in the formation of carbonate radicals that can propagate oxidative damage. The most significant antioxidant role for CO_2_ in vivo is its ability to stabilize the iron–transferrin complex and thereby prevent the involvement of this complex in the initiation of free radical reactions [95].

While the evidence of the relationship between CO_2_ therapy and oxidative stress is limited, Veselá and Wilhelm found that CO_2_ plays a protective role in scavenging free radicals and suppressing oxidative metabolism [95]. A recent study demonstrated that CO_2_ treatment could reduce the level of asymmetric dimethylarginine, which is a marker of oxidative stress [96].

### 3.7. Benefits of Diabetes Mellitus

Microvascular disease caused by diabetes is a potential target for CO_2_ therapy. Studies have shown that the basic status of the microvasculature is one of low O_2_ and low metabolism, although the role of HIF-1α in the tissue response to hyperglycemia and hypoxia appears to be rather variable [97]. HIF-1α plays an unpredictable role in the response of the tissue to hyperglycemia, and current data suggest that it is probably better to activate the CO_2_ hypoxemic response than the HIF-1α hypoxic pathway. Hyperglycemia triggers the glycolytic pathway for ATP production. This pathway generates less CO_2_ compared to aerobic respiration. Therefore, the concentrations of CO_2_ in the tissue are lower than normal. This sets up an environment with low oxygenation, increased inflammation, and reduced capillary density. Notably, exercise can reduce these complications of diabetes [98], and it appears that CO_2_ therapy can mimic these same effects [28] as well as re-balance the pro- and anti-factors controlling angiogenesis, such as VEGF, endothelial NO synthase, and thrombospondin-1 [26].

### 3.8. Reducing Tumor Hypoxia

Tumor growth is limited by the need for O_2_ and the limited diffusion of nutrients. Solid tumors display a hypoxic core that stimulates pathways triggered by HIF-1α [43,45,99]. As noted above, numerous studies have consistently confirmed that CO_2_ therapy will decrease the size of tumors by reducing the hypoxic core of the tumor, reversing the stabilization of HIF-1α to modulate survival pathways, and triggering non-HIF-related pathways [43,45,99]. CO_2_ therapy results in an oxygenated core in the tumor that increases tumor cell apoptosis and enhances sensitivity to chemotherapy and radiation therapy. A collection of studies describes multiple mechanisms that could come together, leading to apoptosis due to the presence of CO_2_ and improved oxygenation. These pathways involve increases in intracellular calcium, AMPK, PGC-1α, and caspases 3 and 9 [28,76,100,101,102,103,104,105].

## 4. Summary

Therapeutic CO_2_ gas has been found to be effective for many microvascular disorders. Figure 2 summarizes some of the tissue responses to hypoxia and elevated CO_2_ and demonstrates how CO_2_ protects the tissue from progressing toward survival mode (e.g., by reducing HIF-1α and activating AMPK) when hypoxia is severe and/or sustained. Elevated CO_2_ activates mechanisms that pull the microenvironment toward a “thrive mode” that maintains homeostasis. Physiological properties triggered by CO_2_ include elevated oxygenation, increased angiogenesis, increased mitochondrial biogenesis, anti-inflammation, reduced oxidative stress, and increased tissue blood flow. The balance between pro-inflammatory and anti-inflammatory phases is also pushed toward homeostasis when tissue CO_2_ is elevated. This is critical for healing wounds and fractures. Hyperglycemia in diabetes leads to poor oxygenation, loss of capillaries, and inflammation. Diabetes is a risk factor for peripheral arterial disease and reduced blood flow, often leading to chronic wounds. Therapeutic CO_2_ corrects this and restores homeostasis. Tumors have a hypoxic core that keeps metastatic cells alive due to pathways that are triggered by HIF-1α. Therapeutic CO_2_ can help oxygenate the hypoxic core and reduce HIF-1α. Finally, evidence supports the thesis that CO_2_ is a critical metabolic factor during exercise that triggers angiogenesis, mitochondrial biogenesis, and myocyte fiber switching. Thus, CO_2_ is not only a metabolic waste product but may be a gaseous signaling molecule, similar to NO, carbon monoxide, and hydrogen sulfide. Modulating its levels via therapeutic application may provide a path to multiple important health solutions.

## Figures and Tables

**Figure 1 ijms-24-05181-f001:**
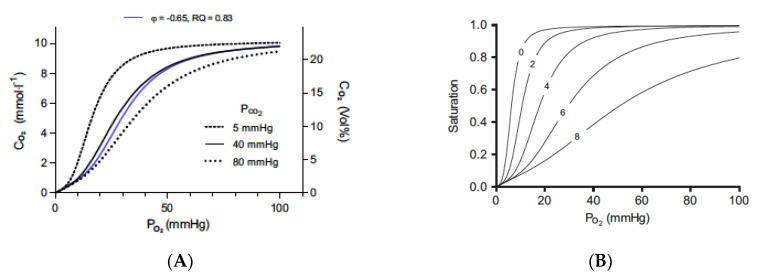
The Bohr effect, as defined by CO_2_ and protons. The Bohr and Haldane effects have been modeled to describe the interactions between O_2_ and CO_2_ binding to hemoglobin [72]. The Bohr effect describes how O_2_ binding is decreased in the presence of CO_2_ and protons. Graph (**A**) shows the traditional oxygen (**A**) equilibrium curves modeled under different levels of CO_2_. The three black curves are the oxygen equilibrium curves obtained when the partial pressure of CO_2_ is held constant at the indicated values. The blue curve is obtained when CO_2_ is added at the same time as oxygen is removed at a gas exchange ratio, respiratory quotient (RQ), of 0.83. Graph (**B**) shows the effect of protons (B) on binding with a constant PCO_2_ value of 40 mmHg and pH 7.2 at a PO_2_ of 100 mmHg. Protons bind to the Bohr groups on the hemoglobin. The five oxygen equilibrium curves are obtained by increasing the number of bound Bohr groups (i.e., protons bound), as indicated by the numbers on the curves. The pH is not displayed because it is a poor indicator of proton binding. As protons are bound to hemoglobin, the pH increases. Figures are reprinted/adapted with permission from Ref. [72]. Copyright year 2018, Journal of Applied Physiology.

**Figure 2 ijms-24-05181-f002:**
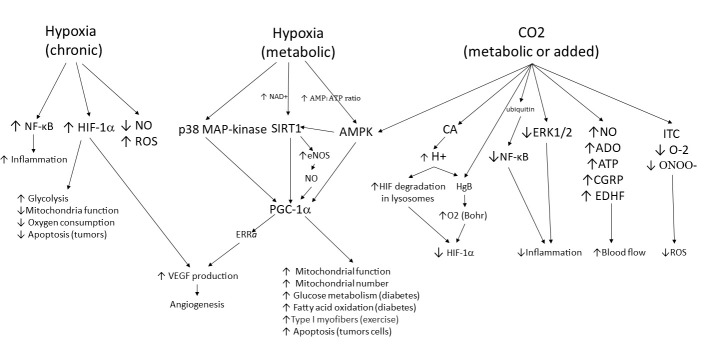
Tissue response to hypoxia and carbon dioxide. This chart shows the molecular pathways that are involved in the tissue response to hypoxia and elevated CO_2_ levels. Three pathways are shown. The first two are tissue responses to hypoxia. Normal metabolic control of hypoxia initially involves the moment-to-moment response to O_2_ demands by alterations in blood flow. If demand persists, pathways for improving O_2_ supply are activated. Multiple pathways lead to the activation of PGC-1α, which induces metabolic changes and new blood vessel formation to allow the tissue to continue to thrive. If hypoxia is persistent and/or severe, as seen in pathological conditions, the left pathway is also activated. Cells experience increased inflammation and oxidative stress and activate HIF-1α-induced production of VEGF to further stimulate neovascularization and increase blood flow while also altering cell metabolism to reduce cell death (i.e., stimulate survival mode). The right pathway shows the responses activated by increases in tissue CO_2_. When blood flow is limited, CO_2_ levels rise and trigger activation of AMPK, stimulating the tissue to respond with changes in cell metabolism and neovascularization that allow the tissue to thrive. Multiple additional pathways are stimulated by elevated CO_2_, resulting in increased blood flow, decreased inflammation, decreased oxidative stress, as well as a reduction in survival pathways stimulated by HIF-1α. Thus, elevated CO_2_ levels (either metabolically generated or brought about by therapeutic CO_2_ delivery) will produce a more homeostatic state in the tissue.

## Data Availability

All data reported is publicly available.

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
