# Peer review of "The Tissue Response to Hypoxia: How Therapeutic Carbon Dioxide Moves the Response toward Homeostasis and Away from Instability"

_ijms, 2023, doi:10.3390/ijms24065181_

Round 1

Reviewer 1 Report

I thank the authors for producing this review which highlights the therapeutic effects of carbon dioxide and explains in depth the physiological mechanisms of action associated with CO2-induced hypoxia. It is a difficult exercise that deserves recognition.
Nevertheless, as it stands, the paper is very unequivocal and presents CO2 a bit as a "miracle cure" that heals wounds, decreases tumors size, reduces inflammation and could replace exercise in diabetics. It lacks a "methodology" part which explains the choice of articles and which would allow the reader to keep in mind the point of view of the authors. CO2 undeniably has positive effects on the tissues but it all depends on the exposure time, the concentration, the number of repetitions, etc... These parameters as well as the risks should be discussed or at least mentioned.

Likewise, it is not always clear when CO2 is given intradermally, in a bath or as breathing gas. Ex: line 155 "Elevating tissue CO2 can cause skeletal muscle to respond as though it has been exercising". Is the increase in CO2 at the muscle level given via a transcutaneous cream or via a breathing gas? Are the effects the same?
-same comment for line 160-161: "These data are consistent with the greater endurance and more efficient muscle activity noted in rats and humans following CO2 therapy". How was CO2 administrated ?
It could be interesting to add a paragraph on the different ways of administering CO2 (cream, bath, gas, ...), in which case it is preferable to use this or that technique and if the effects and/or mechanisms are the same.

Here are some specific comments:
- The introduction does not include any references. Line 42, the authors speak about "well-described pathways". References are therefore necessary here.
- Same as line 45 "Recently, scientific studies have revealed..." References should also be added here.
- line 209: the abbreviation ARDS should be spelled out. Unless I am mistaken, it is not explained above.
- lines 222-224: "Thus, while CO2 treatment alone can decrease the potential size of tumors, [...], the response is even more dramatic. Wouldn't the word "effective" or "greater" be better suited that dramatic when reducing the size of cancerous tumors is precisely the desired effect of chemotherapy or radiation? - line 422: "Some of the beneficial effects of therapeutic CO may be [...]". I assume it is CO2 and not CO.

- Figure 2: Figure 2 is very interesting for paper but could be improved:

(1) An explanatory text of the figure is needed (or adapt the summary) in order to allow the reader to follow the steps of the text directly on figure 2.

(2) what do the dotted lines mean compared to the solid lines? Inhibition versus activation? This should be added in the legend.

(3) I don't quite understand why you note in the hypoxia part an increase in eNOS with an increase in NO (I assume, there is no arrow). Indeed, hypoxia has been shown to reduce FMD (Theunissen, 2022). Hypoxia also increases the expression and activity of Rho-kinase (Takemoto, 2002). Rho kinase facilitates the downregulation of the eNOS expression induced by hypoxia (Takemoto, 2002), thus reducing the endothelial production of NO. 

(4) NFkB should also appear in the hypoxia part. Indeed, "Oxygen deprivation results in a significant increase in the expression of iNOS due to the activation of the nuclear transcription factor kappa B (NF-κB) (Jiang, 2009; Moro, 1998). The effects of hypoxia on cancer cells are opposed to those of CO2 since hypoxia activates HIF-1alpha and NF-κB. Hypoxia decreases apoptosis (tumors) via HIF-1a but increases apoptosis (tumors) via PGC-1a. The literature seems to agree that hypoxia is a driver of Cancer Chronic Inflammation through HIF-1 and NF-κB Activation (Korbecki, 2021). This is not so clear in the figure.

(5) ROS and ONOO- should also appear in the hypoxia part.

(7) It is not very clear if the hypoxia part is the one created by the pathologies (tumor, diabetes, exercise, etc...) in the tissues or created by the organism in response to the CO2 therapy.

I really encourage the authors to improve the summary figure and to add a text legend to help the reader understand the complex mechanisms involved in hypoxia and CO2 therapy.

Jiang ZL, Fletcher NM, Diamond MP, Abu-Soud HM, Saed GM. Hypoxia regulates iNOS expression in human normal peritoneal and adhesion fibroblasts through nuclear factor kappa B activation mechanism. Fertil Steril. 2009 Feb;91(2):616-21. doi: 10.1016/j.fertnstert.2007.11.059. Epub 2008 Feb 20. PMID: 18281043; PMCID: PMC2812021.

Korbecki J, SimiÅ„ska D, GÄ…ssowska-Dobrowolska M, Listos J, Gutowska I, Chlubek D, Baranowska-Bosiacka I. Chronic and Cycling Hypoxia: Drivers of Cancer Chronic Inflammation through HIF-1 and NF-κB Activation: A Review of the Molecular Mechanisms. Int J Mol Sci. 2021 Oct 2;22(19):10701. doi: 10.3390/ijms221910701. PMID: 34639040; PMCID: PMC8509318.

Moro MA, De Alba J, Leza JC, Lorenzo P, Fernández AP, Bentura ML, Boscá L, Rodrigo J, Lizasoain I. Neuronal expression of inducible nitric oxide synthase after oxygen and glucose deprivation in rat forebrain slices. Eur J Neurosci. 1998 Feb;10(2):445-56. doi: 10.1046/j.1460-9568.1998.00028.x. PMID: 9749707.

Takemoto M, Sun J, Hiroki J, Shimokawa H, Liao JK. Rho-kinase mediates hypoxia-induced downregulation of endothelial nitric oxide synthase. Circulation. 2002 Jul 2;106(1):57-62. doi: 10.1161/01.cir.0000020682.73694.ab. PMID: 12093770.  

Theunissen S, Balestra C, Bolognési S, Borgers G, Vissenaeken D, Obeid G, Germonpré P, Honoré PM, De Bels D. Effects of Acute Hypobaric Hypoxia Exposure on Cardiovascular Function in Unacclimatized Healthy Subjects: A "Rapid Ascent" Hypobaric Chamber Study. Int J Environ Res Public Health. 2022 Apr 28;19(9):5394. doi: 10.3390/ijerph19095394. PMID: 35564787; PMCID: PMC9102089.

Reviewer 2 Report

This is an interesting review of the anti-hypoxic effect of CO2 in the body. The Authors first summarize the clinical instances where CO2 treatments were revealed to be beneficial, then try to give a molecular-based justification on how CO2 affects some of the biochemical paths that lead to protection. A few issues, however, require some attention.

The concept of the thrive vs survival mode is intriguing but not developed enough. 

Introduction. Although the vasodilation effect of CO2 is generally accepted, this matter is still somewhat contradictory. I suggest introducing this story with more impartiality, providing the reasons why CO2 is claimed also to have vasoconstrictor effects. Note that no bibliographic references are cited in the introduction. Please add references to the main studies that developed this theory and the limitations of the vasodilatory effects of CO2, for example, the vascular tone. 

As outlined by the Authors in several, but not all due places, the effect of CO2 is intimately linked to that driven by the proton released during CO2 hydration. Virtually, most if not all the reported mechanisms can be attributed to H+ rather than CO2. The Authors should make an effort to distinguish the effect driven by CO2 itself (at constant pH) from that driven by H+, or at least they should give their opinion (this is allowed in the reviews). 

Line 238. I understand that much is still to be discovered, but the Authors should at least suggest how CO2 interacts with eNOS. This may be one of the instances when small local pH changes due to CO2 may make the difference. Also, the modulation of KATP channels may be quite dependent on H+ because acidosis is known to lead to hyperkalemia. 

Carbonic anhydrase activity may be more critical than expected because it modulates the kinetic of CO2-induced changes in pH.

Paragraph starting at line 316. The interconversion of CO2 to bicarbonate occurs not only in the RBC but also in the plasma. Carbonic anhydrase is needed to accelerate the passage of one-carbon moieties between the two compartments because membranes are permeable to CO2 but not to bicarbonate. The sentence “In the absence of CO2 and protons, hemoglobin will hang on to the O2 until the tissue is essentially anoxic” is not clear. The rest of the paragraph concerns an interesting interpretation of the O2 delivery in the microcirculation. However, as rightly pointed out, the effect of pH changes may dominate over the effect of CO2. The Authors may compare the P50 change driven by a given change in PCO2 at constant pH with that driven by the change in pH corresponding to that change in PCO2. 

Paragraph starting at line 349. The decrease in HIF may be simply due to increased O2 delivered to tissues from Hb due to the Hb decreased O2 affinity, in turn, due to the acidosis led by high PCO2. 

Line 422, is it CO2 or CO? 

Line 440, is the CO2 action as a scavenger of peroxynitrite documented? 

Line 435, replace diabetic with diabetes.

Reviewer 3 Report

The reviewed manuscript entitled ‘The tissue response to hypoxia: how therapeutic carbon dioxide moves the response toward homeostasis and away from instability’ written by Richard J. Rivers and Cynthia J. Meininger gathers very interesting information about biological action of CO2 and therapeutic opportunities. The authors described molecular mechanisms involved in the biological effects exerted by CO2, as well as the potential use of CO2 administration in the therapy of diseases associated with hypoxic conditions. The review is clear, comprehensive, and relevant to the field. Figures and cited references are appropriate. The one aspect that could be improved is the legend to Figure 2, where the explained abbreviations could be ordered alphabetically to facilitate their finding by the readers. It could be done during the proofreading stage.

Reviewer 4 Report

The review by Rivers and Meininger is devoted to the analysis of current literature data on the therapeutic effect of CO2. The review discusses the effects of CO2 on various pathologies and diseases (wound and fracture, ischemia, diabetes mellitus, inflammation, tumor growth and metastasis), and also discusses possible mechanisms of CO2 action. This interesting review is well structured and well written.

Comments:

1.      The number of references in the list since 2018 is about 25%. References should be up-to-date, i.e., 50% or above are the papers published within recent 5 years.

2.      The first part in the title of the review (The tissue response to hypoxia) can be deleted, since it does not fully reflect the meaning of the review.

Round 2

Reviewer 1 Report

I thank the authors for these major corrections that greatly clarify the paper. For my part, I have no further comments and accept the paper in present form.

Author Response

Thank you for taking time to review this paper and help make it better.

Reviewer 2 Report

The Authors have done an excellent job of meeting all my concerns.

As for the first point (vasoconstrictor effects of CO2), I meant exactly pulmonary vasoconstriction. I think such a valuable review may be worth a few sentences describing that CO2 has vasodilating effects in all the body except the pulmonary system, possibly accompanied by a mechanistic explanation for this dual behavior.  

As for the Hb Bohr effect that appears more marked for pH rather than PCO2 changes, the Authors may notice that while PCO2 is expressed in linear terms (a 10%, say, decrease in CO2 molarity corresponds to the same decrease in PCO2), pH is instead expressed in logarithmic terms (at pH 7.4, decreasing the H+ molar concentrations by 10% changes the pH to 7.45 or so).  

Author Response

Thank you for your comments. We have added a paragraph defining the special effects of CO2 in the pulmonary network. We added four more references. The changes are at line 327. 

"

It should be noted that all the aforementioned mechanisms for CO2-dependent increases in blood flow are not relevant in the pulmonary circulation. The pulmonary circulation is unique. The role for CO2 in controlling the distribution of pulmonary blood flow is small compared to the powerful effects hypoxia and pulmonary vasoconstriction [66]. Data shows CO2 can induce both vasodilation and vasoconstriction and it appears to depend on the level of resting vascular tone [67,68]. In sharp contrast to the peripheral vasculature where protons cause vasodilation, CO2 and the CA-dependent production of carbonic acid causes vasconstriction in the pulmonary vascular network [69].  "